# The Intracellular Symbiont *Wolbachia pipientis* Enhances Recombination in a Dose-Dependent Manner

**DOI:** 10.3390/insects11050284

**Published:** 2020-05-06

**Authors:** Kaeli N. Bryant, Irene L. G. Newton

**Affiliations:** Department of Biology, Indiana University, 1001 E 3rd Street, Bloomington, IN 47405, USA; kaelbrya@iu.edu

**Keywords:** *Wolbachia*, recombination, *Drosophila*

## Abstract

*Wolbachia pipientis* is an intracellular alphaproteobacterium that infects 40%–60% of insect species and is well known for host reproductive manipulations. Although *Wolbachia* are primarily maternally transmitted, evidence of horizontal transmission can be found in incongruent host–symbiont phylogenies and recent acquisitions of the same *Wolbachia* strain by distantly related species. Parasitoids and predator–prey interactions may indeed facilitate the transfer of *Wolbachia* between insect lineages, but it is likely that *Wolbachia* are acquired via introgression in many cases. Many hypotheses exist to explain *Wolbachia* prevalence and penetrance, such as nutritional supplementation, protection from parasites, protection from viruses, or direct reproductive parasitism. Using classical genetics, we show that *Wolbachia* increase recombination in infected lineages across two genomic intervals. This increase in recombination is titer-dependent as the *w*MelPop variant, which infects at higher load in *Drosophila melanogaster*, increases recombination 5% more than the *w*Mel variant. In addition, we also show that *Spiroplasma poulsonii,* another bacterial intracellular symbiont of *D. melanogaster*, does not induce an increase in recombination. Our results suggest that *Wolbachia* infection specifically alters its host’s recombination landscape in a dose-dependent manner.

## 1. Introduction

Recombination, the exchange of genetic material during meiosis, is thought to be largely beneficial, as it increases the efficacy of natural selection [1,2]. Because of chromosome architecture, loci that are physically linked to each other can interfere with selection such that selection at one locus reduces the effective population size, and therefore the efficacy of selection, at linked loci. This phenomenon, termed “Hill–Robertson interference,” means that positive or negative selection at one site can interfere with selection at another site. By allowing loci to shuffle between chromosomes, recombination mitigates Hill–Robertson interference [3]. As a result of this re-shuffling, areas of the genome subject to high recombination rates show higher nucleotide diversity, either because of the inherent mutagenic effect of recombination, or by the indirect influence of recombination on natural selection in a population. Overall, a large body of literature supports the assertion that recombination increases efficacy of selection and enhances adaptation in animals, as reviewed and studied in various *Drosophila* species [1,2,3].

One factor that may influence recombination is bacterial infection. For example, injection of *Drosophila melanogaster* with the bacterial pathogen *Serratia* increases recombination post infection [4]. Many *Drosophila* species are colonized persistently by *Wolbachia pipientis*, an alpha-proteobacterium within the *Rickettsiales* and a highly prevalent infection, found in 40%–60% of all insects [5,6]. *Wolbachia’s* prevalence in populations is likely modulated by reproductive manipulations, induced to benefit infected females [7]. However, this reproductive parasitism alone is not sufficient to explain the prevalence of *Wolbachia* infection. Indeed, there are many recently discovered, insect-infecting strains that do not seem to induce any reproductive phenotype at all, suggesting other potential benefits provided by the symbiont [8,9,10]. One known benefit is pathogen blocking, where *Wolbachia* repress virus replication within the insect host [11,12,13]. This phenomenon has important implications for the use of *Wolbachia* in vector control [14]. In addition to protecting their hosts from pathogens, *Wolbachia* also improve the fitness and fecundity of some hosts (by protecting against virus infection or enhancing survival and fecundity), and removal of the endosymbiont can cause a decrease in host fitness [15,16]. Finally, *Wolbachia* can rescue oogenesis defects in mutant *Drosophila* strains [17] and have also made themselves a necessary component of oogenesis in some wasp species, thereby making the infection indispensable [18]. 

One recently discovered phenotype of *Wolbachia pipientis* is that it may increase the frequency of recombination events on the X chromosome, but not on the third, in *D. melanogaster* [19,20]. This phenotype contrasts with that observed for *Serratia* infection, where elevated recombination was observed on the third chromosome [4]. Does *Wolbachia* infection actually lead to increased recombination? If so, would any infection of the reproductive tract result in increased recombination? Here, we answer these questions using classical genetics in *D. melanogaster* with different *Wolbachia* variants and using another maternally transmitted bacterial infection, *Spiroplasma poulsonii.* We confirm that *Wolbachia* significantly increase recombination across two intervals, one on the X and one on the second chromosome, but we could not detect any effect on the third chromosome interval queried. In addition, there is a clear correlation between *Wolbachia* load and recombination events, suggesting *Wolbachia* infection itself is the cause of the elevated recombination; clearing the host of *Wolbachia* restores recombination rate to a basal level while infection with a high-titer variant, in the same genetic background, increases recombination. Another intracellular symbiont, *Spiroplasma,* does not increase recombination rate, suggesting this phenomenon is not simply due to the presence of a bacterial infection in the gonads, but is induced by *Wolbachia*. These results, in sum, confirm that *Wolbachia* infection elevates host recombination.

## 2. Materials and Methods

### 2.1. Fly Rearing

Flies were ordered from the Bloomington Drosophila Stock Center. Three marker stocks were selected as recombination trackers for the X, 2nd, and 3rd chromosomes in *D. melanogaster*: #1509, which is marked with *yellow* (*y*) and *vermillion* (*v*) on the X chromosome (33.0 cM); #433, which is marked with *vestigial wings* (*vg*) and *brown* (*bw*) on the 2nd chromosome (37.5 cM); and #496, which is marked with *ebony* (*e*) and *rough* (*ro*) on the 3rd chromosome (20.4 cM). Two stocks, one *Wolbachia* infected and one uninfected, were selected at random from the Drosophila Genetic Reference Panel (DGRP): DGRP-320 and DGRP-83, respectively. To modulate infection status, we introduced high-titer *Wolbachia* infections into our stock DGRP-320 using the *Wolbachia* variant *w*MelPop from stock BDSC #65284. To clear DGRP-320 of its infection, flies were raised on fly food containing 50 ug/mL tetracycline for 3 generations and then allowed to be recolonized by their extracellular microbiome, and recover from tetracycline, for 1 generation. All crosses were conducted at room temperature.

### 2.2. Spiroplasma Methods

An original *Spiroplasma*-infected, wild-caught *D. melanogaster* line was a generous gift from John Jaenike. Hemolymph from 1-month-old *Spiroplasma*-infected female flies, which are heavily colonized by the symbiont [21], was suspended in sterile phosphate buffered saline (PBS), then injected intrathoracically into virgin uninfected OreR-modENCODE (*Wolbachia*-negative) females with a microinjector (*n* = 67). Flies were allowed to recover in individual vials for 3 days prior to adding 3—5 uninfected OreR-modENCODE males. A successful infection was identified first by observing sex ratios of progeny, then verified by PCR using the PCR protocol below. *Spiroplasma*-infected lines are maintained on standard Bloomington food at 25 °C on a 12 h light/dark cycle with an addition of OreR-modENCODE uninfected males each generation. 

### 2.3. DNA Extraction and Polymerase Chain Reaction

*Wolbachia* and *Spiroplasma* infection across all flies was confirmed by PCR, in triplicate reactions with no template controls. DNA was extracted using a single-fly extraction method. Whole flies were ground with a pipette tip containing 50 μL lysis buffer (10 mM Tris-HCl pH 8.2, 1 mM EDTA, and 25 mM NaCl) and 5 μL Proteinase K. They were incubated at room temperature for 20 min then heated to 95 °C for 2 min to deactivate the enzyme. After a quick spin, this lysate was directly used as a template for PCR. For *Wolbachia,* primers used for this are as follows: wsp F1 5′-GTC CAA TAR STG ATG ARG AAA C-3′ and wsp R1 5′-CYG CAC CAA YAG YRC TRT AAA-3′. The cycling conditions were: 98 °C for 2 min, followed by 30 cycles of 98 °C for 30 s, 59 °C for 45 s, and 72 °C for 1 min 30 s, then finished with 72 °C for 10 min. For *Spiroplasma*, primers used were: SpiroDnaA_F 5′–TTAAGAGCAGTTTCAAAATCGGG–3′ and SpiroDnaA_R 5′–TGAAAAAAACAAACAACAAATTGTTATTACTTC—3’ [22,23]. The cycling conditions were: 98 °C for 2 min, followed by 35 cycles of 98 °C for 30 s, 56.7 °C for 1 min, and 72 °C for 30 s, then finished with 72 °C for 5 min. Amplified DNA was visualized using agarose gel electrophoresis. Quantitative PCR was performed to confirm the titer difference in *Wolbachia* infection between *w*Mel and *w*MelPop. Data were collected using an Applied Biosystems StepOne Real-time PCR system and iTaq universal SYBR Green supermix. The *Wolbachia* primers used are as follows: wspF 5′-CATTGGTGTTGGTGTTGGTG-3′ and wspR 5′-ACCGAAATAACGAGCTCCAG-3′. The host primers used are as follows: Rpl32F 5′-CCGCTTCAAGGGACAGTATC-3′ and Rpl32R 5′-CAATCTCCTTGCGCTTCTTG-3′. The cycling conditions are as follows: 50 °C for 2 min, 95 °C for 10 min, followed by 40 cycles of 95 °C for 30 s and 59 °C for 1 min. The reaction was carried out in a 96-well plate. Gene expression was determined by the Livak and Pfaffl methods. 

### 2.4. Recombination Assay

To determine if recombination events had occurred, a two-step crossing method was devised, shown in Figure 1. Ten virgin DGRP females aged 3–5 days were housed with ten phenotypically marked males and were allowed to mate until the appearance of pupae, after which the parents were cleared from the bottle. Virgin female F1 progeny was collected and crossed to the male parental line in the same ratio as before and allowed to mate for 10 days before being cleared from the bottle. All F2 progeny from this cross was collected and frozen within 10 days of clearing the parental F1s. The flies were scored according to their visible phenotypes and sorted into two groups. The proportion of recombinants in each cross was calculated by adding the total exhibiting visible phenotypes and dividing by the total number of progeny. Differences in these proportions between *Wolbachia* infected and uninfected flies were tested using a logistic regression and Mann–Whitney U tests in SPSS.

## 3. Results

### 3.1. Wolbachia Infection Increases Host Recombination Rate

Prior work has shown that *Wolbachia* infection is correlated with an increased frequency of recombination in natural populations [19,20]. We used the same set of isogenized flies, sampled from a wild-caught population in North Carolina, the Drosophila Genetic Reference Panel [24], to confirm prior observations of increased recombination in a *Wolbachia*-infected fly. Virgin females from two backgrounds (DGRP-320, infected with *Wolbachia*, and DGRP-83, *Wolbachia-*free) were crossed independently to three different lines carrying chromosomal markers, allowing us to distinguish recombinants along certain genomic intervals based on the presence of dominant markers (Figure 1). For the X and second chromosomes, we observed an increase in recombination rate of 6.4% and 6.1%, respectively, based on *Wolbachia* infection status (df = 1, χ^2^ = 14.316, *p* < 0.001 and df = 1, χ^2^ = 5.597, *p* = 0.018, for the X and second chromosome data) (Figure 2). This observation is well within the range of what had previously been observed for these same genotypes [19]. No statistically significant effect of *Wolbachia* infection was observed on the third chromosome (df = 1, χ^2^ = 2.156, *p* = 0.142). It may be that this difference in recombination rate across chromosomes reflects the natural variation in recombination observed across genomic intervals for *D. melanogaster,* or it may be an artifact of the genomic interval sampled and not an influence of *Wolbachia* on specific chromosomes [19]. 

Although this first experiment established we could replicate prior results, it did not control for host genetic background. To control for host genotype, we cleared the *Wolbachia* infection from two fly stocks by rearing the flies on tetracycline for three generations and then repopulating the extracellular microbiome for one generation (Appendix A). We performed genetic crosses using these cleared backgrounds, comparing them to the uncleared progenitor lines in each case, focusing on the X chromosome interval as we had already established that *Wolbachia* significantly increased recombination across that genomic interval. For both of these fly stocks, we observed an increase in recombination on the X chromosome when flies were infected with *Wolbachia* (Figure 3A; df = 1, χ^2^ = 16.363, *p* < 0.001 for the OreR-modENCODE background and df = 1, χ^2^ = 4.461, *p* = 0.035 for OreR-C). Specifically, we observed an increase in recombination of 6.4%; this increase was well within the range we had observed in our first experiments (Figure 2).

### 3.2. Dose-Dependent Effect of Wolbachia on Host Recombination Rate

Because we observed an influence of *Wolbachia* infection status on the host recombination rate, we sought to modulate infection status by using high-titer *Wolbachia* infections in our experiment. *Wolbachia* colonize *D. melanogaster* at different titers depending on the amplification of a specific genomic interval in the *Wolbachia* genome termed “octomom” [25]. We crossed females carrying the highest-titer, pathogenic *Wolbachia* variant, *w*MelPop, (w[1118]/Dp(1;Y)y[+] |Wolbachia-wMelpop; BDSC stock #65284), with males of stock DGRP-320. The lines were introgressed for three generations within the #DGRP-320 genetic background before use in experiments (Appendix A). We also cleared the *Wolbachia* infection from the DGRP-320 line, using the same tetracycline method as noted above. We looked specifically at the X chromosome interval. Again, we observed a significant effect of *Wolbachia* infection on recombination rate in this experiment (df = 3, χ^2^ = 33.4, *p* < 0.001) (Figure 3). Interestingly, we observed a significant effect of *Wolbachia* titer on recombination rate—the high-titer *w*MelPop variant increased recombination on the X chromosome in F2 progeny by 9.5% compared to the 6.3% observed for *w*Mel (df = 3, χ^2^ = 4.866, *p* = 0.027) (Figure 3). 

### 3.3. Spiroplasma Does Not Increase Host Recombination Rate

We hypothesized that *Wolbachia* may be a stress on the host cell, increasing recombination rate as a result of increased reactive oxygen species or other immune activation pathways. We therefore reasoned that any bacterial infection may increase recombination rate. To test this hypothesis, we procured *Spiroplasma poulsonii* MSRO (a gift from John Jaenike), which we used to infect a *Wolbachia-*free OreR lab stock (Oregon-R-modENCODE, BDSC #25211) (Appendix A). We used the same crossing scheme as above to introduce *Spiroplasma* into the y[1] v[1] background (Figure 1), carrying phenotypic markers on the X chromosome. As a genetic control, we used stock #25211. Counter to our hypothesis, we observed no significant increase in recombination based on *Spiroplasma* infection (Figure 4 df = 1, χ^2^ = 0.057, *p* = 0.812). 

## 4. Discussion

For sexually reproducing organisms, recombination is both a source of genetic diversity within a population and a mechanism by which to decouple differential selection on sites across the chromosome. Therefore, recombination is thought to be beneficial. Here, we observed that *Wolbachia* infection significantly increased the recombination rate observed across two genomic intervals (for both the X and the second chromosomes). Our goal was to identify whether the *Wolbachia* infection itself was responsible for the increase in recombination rate that was previously observed. We showed that it is because an increase in *Wolbachia* titer increases the recombination rate proportionally and because another endosymbiont, *Spiroplasma*, does not elevate recombination rate. Recombination rates vary dramatically across animals, even within a genus, as best illustrated within the *Drosophila* clade [26,27]. The mechanism behind this difference is not well understood, but our data suggest that the symbiont *Wolbachia* may influence the recombination rate of infected *D. melanogaster*. 

The mechanism by which *Wolbachia* infection elevates recombination is not currently known. *Wolbachia* have an active type IV secretion system that they use to secrete proteins into the host and modulate host cell biology. It is possible that some of these proteins may influence the recombination rate directly or indirectly, although no effectors have yet been identified that bind to host DNA. Importantly, some of the recently identified and characterized cytoplasmic incompatibility (CI) factors contain nuclease domains [28], and for the *cin* loci, nuclease activity against DNA in vitro has been directly shown [29]. *Wolbachia* also encode many other secreted effectors that might directly interact with host proteins involved in recombination [30]. It is not clear how an increase in double stranded breaks would increase the recombination rate, nor what host loci might modulate this increase in recombination. That said, variation in the recombination rate across *Drosophila melanogaster* genotypes has been clearly observed [19], and a similar approach, modulating *Wolbachia* infection status in each of these backgrounds, could identify host loci that modulate the recombination phenotype we observe. 

Here, we used two different *Wolbachia* variants and multiple host genotypes to support the hypothesis that *Wolbachia* increases host recombination rate. However, it is possible that strains outside of the *w*Mel clade do not increase the host recombination and a comparative genomic framework could be used to identify loci in *Wolbachia* that confer the phenotype. *Wolbachia pipientis* is certainly not monolithic, and genome content differs significantly across the genus [31]. The clearest comparison would come from using closely related strains, in the same host *D. melanogaster* background, identifying loss of recombination rate increase. As titer is a confounding variable, this must be controlled in the experiment. Finally, a recent publication suggested *Wolbachia w*Mel-infected *Drosophila* prefer cooler temperatures [32]. Increases in temperature modulate recombination in *D. melanogaster* [33], and it is possible that *Wolbachia* infection elevates host temperature enough to generate an increase in the number of detected recombinants, in a laboratory setting where flies are kept at a constant temperature. 

## 5. Conclusions

In this study, we used genetic crosses and varying titers of *Wolbachia pipientis* to support the hypothesis that *Wolbachia* infection increases recombination rate in *Drosophila melanogaster.* Because *Spiroplasma* infection does not induce a similar phenotype, we hypothesize that *Wolbachia pipientis* specifically upregulates recombination, in a dose-dependent manner and that all bacterial infections do not increase recombination rate *per se.* The mechanism by which this increase in recombination is achieved is not known but we hypothesize that some manipulation of recombination machinery (expression, localization, etc) coupled to DNA damage might induce the phenotype observed.

## Figures and Tables

**Figure 1 insects-11-00284-f001:**
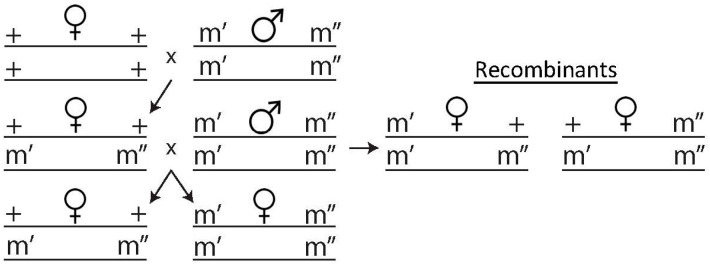
Schematic of fly crossing used to track recombination events. ++ refers to wild type and m’ m” refers to genetic markers on each chromosome (*y v* on the X; *vg bw* on the 2nd; *e ro* on the 3rd). Recombination is tracked by looking at the ratio of recombinants to the total number of progeny produced.

**Figure 2 insects-11-00284-f002:**
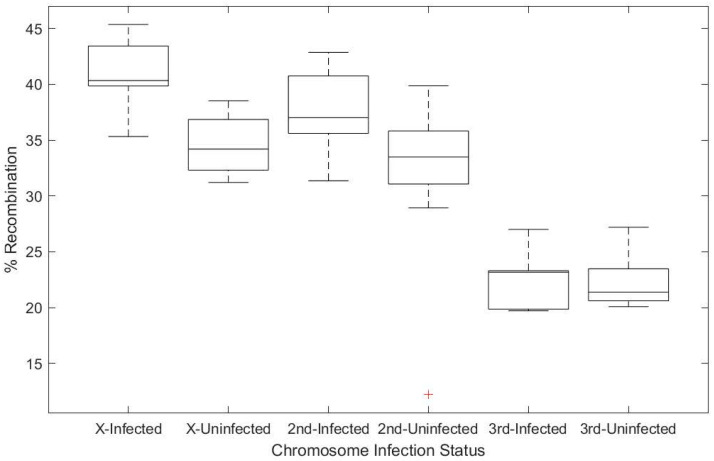
Box and whisker plots showing percentage of recombinants observed, based on genetic markers on the X, 2nd, and 3rd chromosome of *Drosophila*. *Wolbachia* infection significantly increased the recombination rate observed on both the X and 2nd chromosome by 6.4 and 6.1%, respectively. A significant increase was observed for the X (df = 1, χ^2^ = 14.316, *p* < 0.001) and the 2nd chromosome (df = 1, χ^2^ = 5.597, *p* = 0.018), but not on the 3rd chromosome (df = 1, χ^2^ = 2.156, *p* = 0.142). Uninfected = DGRP-83; Infected =DGRP-320.

**Figure 3 insects-11-00284-f003:**
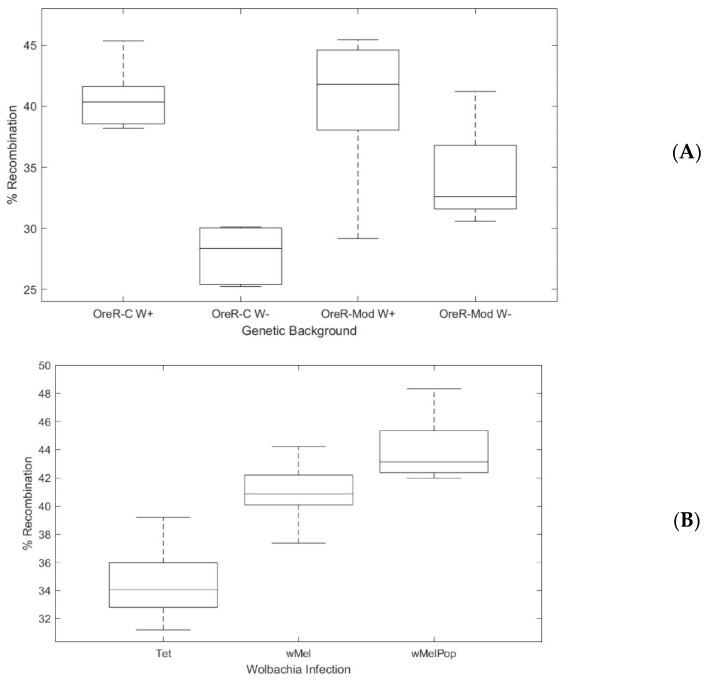
*Wolbachia* increase in recombination is titer depedent and independent of fly genetic background. (**A**). Percentage of recombinants observed based on genetic markers on the X chromosome of *Drosophila* in two different genetic backgrounds, OreR-modENCODE (BDSC stock #25211)and OreR-C (BDSC stock #5), shown as box and whisker plots. *Wolbachia* infection significantly increased the recombination rate observed in both genetic backgrounds. Significance was determined by logistic regression OreR-modENCODE (df = 1, χ^2^ = 16.363, *p* < 0.001) and OreR-C (df = 1, χ^2^ = 4.461, *p* = 0.035). (**B**). Box and whisker plots showing percentage of recombinants observed, based on genetic markers on the X chromosome of *Drosophila*, when *Wolbachia* titer is varied. A uniform genetic background was used for comparisons across *Wolbachia*-uninfected (Tet) flies, *Wolbachia* infected (*w*Mel) flies, and flies infected with a high-titer variant (*w*MelPop) (see Appendix A). Increased *Wolbachia* increased recombination in a load-dependent manner by 6.3% for *w*Mel and 9.5% for *w*MelPop (df = 3, χ^2^ = 33.4, *p* < 0.001). Significance between Tet and *w*Mel (U = 1.0, Z = −3.064, *p* = 0.002), *w*Mel and *w*MelPop (U = 8, Z = −2.870, *p* = 0.004), and Tet and *w*MelPop (U = 0, Z = −3.182, *p* = 0.001) was determined with Mann-Whitney U test implemented in SPSS.

**Figure 4 insects-11-00284-f004:**
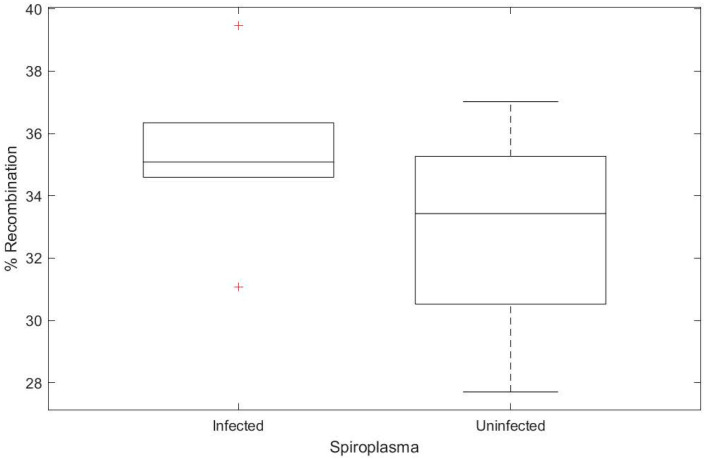
*Spiroplasma* infection does not increase the percentage of recombinants observed based on genetic markers on the X chromosome of *Drosophila*. Box and whisker plots showing that *Spiroplasma* infection did not significantly increase the observed recombination rate (df = 1, χ^2^ = 0.057, *p* = 0.812). Genetic background for all flies was OreR-modENCODE.

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
