# Peer review of "The Intracellular Symbiont Wolbachia pipientis Enhances Recombination in a Dose-Dependent Manner"

_insects, 2020, doi:10.3390/insects11050284_

Round 1
Reviewer 1 Report
Using classical genetics of Drosophila, the authors confirmed the recent results by Nadia Singh that Wolbachia increase recombination in Wolbachia-infected flies. They also show that this increase is likely to be titer-dependent because heavily infected flies increases recombination compared to the normally infected flies. Interestingly, they also show that Spiroplasma, another reproductive symbiont, does not increase recombination. This manuscript is well written and results are clear and significant. I do not have any concern about the publication of this paper. I have only minor comments below. - Line 118: “All F2 progeny from this cross were collected and frozen after 10 days of the clearing.” It is not clear to me how the all F2 progeny were collected. Their parents (F1) were allowed to mate for 10 days. If you try to collect all the F2 progeny from the vial, F3 progeny will definitely be included. Please make it clear in the manuscript if necessary. - In the last sentence of Discussion (Line 246), the authors noted “Wolbachia may also increase introgression”. It is interesting, but I cannot imagine how they can do it. Do you have any idea how Wolbachia can increase introgression?Author Response
Please see attachment.

Reviewer 2 Report
The authors present a short but very interesting and well written study providing evidence that the endosymbiont Wolbachia increases recombination based on bacterial load. The manuscript is well written, and the conclusions are mostly appropriate for the results obtained. I have some minor comments as follows that I think will improve the quality of the manuscript:
Abstract
Line 8-9: Wolbachia ‘is’ in first sentence, Wolbachia ‘are’ in second sentence so consistency needed.
Line 19: the ‘other’ would imply that there are no other intracellular symbionts, so this is quite a strong statement. Would ‘another’ rather than ‘the other’ be more appropriate here?
Introduction
Lines 35-37: ‘a large body of literature’ does not match up with only three references. Suggest ‘reviewed in’ or include more refs
Line 38: flies are a large group of insects – clarify with fruit flies or Drosophila etc
Line 41: ‘infection on the planet’ is a term I personally would not use and need a ref for 40-60% infection rates
Line 49: ‘generally improves’ is ambiguous
Line 58: abbreviate Drosophila (D.) melanogaster and use abbreviation D. from that point on.
Line 65: repetitive use of ‘intracellular symbiont’
Materials and Methods
Line 83: some rationale for using 1 month old flies for hemolymph transfer I think is needed. Is this due to build up of spiroplasma density over time in adult flies?
Line 94: please check journal requirements but spelling out ‘microliters’ doesn’t seem correct
Line 96: was the supernatant not removed to prevent debris from entering PCR reactions?
Line 111 – I assume performed in triplicate with NTCs etc? I would add these details here.
Results
Line 131 – 137: this is really methods so I would reduce as I understand your trying to set out what crosses were being used.
Line 144: 3rd chromosome stats needed here (despite being no different)
Line 145-148: these sentences would seem to fit better in a discussion so consider moving
Line 148 – ref comes after full stop/period
Line 157: Fig 3 appears in text before Fig 2 (line 160)?
Line 163, 183 and 208 (Figure legends): Assume these are all box-whisker plots but this is not mentioned in the legend. Also, would it not be useful to add the P values between selected important comparisons onto the figures?
Discussion
Line 218: the idea of it being ‘Wolbachia-specific’ needs some caveats given only one additional bacterial spp (Spiroplasma) was tested.
Lines 224- 225: ‘active area of enquiry in our lab’ is strange to include in a manuscript
Line 225: Wolbachia ‘have’…again consistency of singular or plural for Wolbachia
Lines 238: ‘known for it’s ability to transfer through species’ is a strange sentence that needs re-writing
Lines 243: the wAu strain does not induce CI but persists in populations so a discussion on how this strain fits in here would be good.
Lines 255 onwards (References)
Some references appear inconsistent in style (eg. 4, 16)
